# Longitudinal Assessment of Multiple Immunological and Inflammatory Parameters during Successful DAA Therapy in HCV Monoinfected and HIV/HCV Coinfected Subjects

**DOI:** 10.3390/ijms231911936

**Published:** 2022-10-08

**Authors:** Paola Zuccalà, Tiziana Latronico, Raffaella Marocco, Stefano Savinelli, Serena Vita, Fabio Mengoni, Tiziana Tieghi, Cosmo Borgo, Blerta Kertusha, Anna Carraro, Gabriella D’Ettorre, Vincenzo Vullo, Claudio Maria Mastroianni, Grazia Maria Liuzzi, Miriam Lichtner

**Affiliations:** 1Department of Public Health and Infectious Diseases, Sapienza University of Rome, 00185 Rome, Italy; 2Department of Biosciences, Biotechnologies and Biopharmaceutics, University of Bari, 70126 Bari, Italy; 3Infectious Diseases Unit, Sapienza University of Rome, Polo Pontino, S. M. Goretti Hospital, 04100 Latina, Italy; 4Department of Infectious Diseases, St. Vincent’s University Hospital, D04 YN26 Dublin, Ireland; 5Department of Neuroscience, Mental Health and Sense Organs (NESMOS), Sapienza University of Rome, 00185 Rome, Italy

**Keywords:** HCV, HIV/HCV, direct-acting antiviral, IP-10, sCD163, sCD14, MMP-2, monocytes, CD4+ T-cells, CD8+ T-cells, sDC

## Abstract

In the direct-acting antiviral (DAA) era, it is important to understand the immunological changes after HCV eradication in HCV monoinfected (mHCV) and in HIV/HCV coinfected (HIV/HCV) patients. In this study, we analyzed sub-populations of monocytes, dendritic cells (DCs), T-lymphocytes and inflammatory biomarkers following initiation of DAA in 15 mHCV and 16 HIV/HCV patients on effective antiretroviral therapy at baseline and after sustained virological response at 12 weeks (SVR12). Fifteen age- and sex-matched healthy donors (HD) were enrolled as a control group. Activated CD4+ and CD8+ T-lymphocytes, mDCs, pDCs, MDC8 and classical, non-classical and intermediate monocytes were detected using flow cytometry. IP-10, sCD163 and sCD14 were assessed by ELISA while matrix metalloproteinase-2 (MMP-2) was measured by zymography. At baseline, increased levels of IP-10, sCD163 and MMP-2 were found in both HIV/HCV and mHCV patients compared to HD, whereas sCD14 increased only in HIV/HCV patients. After therapy, IP-10, sCD163 and sCD14 decreased, whereas MMP-2 persistently elevated. At baseline, activated CD8+ T-cells were high in HIV/HCV and mHCV patients compared to HD, with a decrease at SVR12 only in HIV/HCV patients. Activated CD4+ T-cells were higher in HIV/HCV patients without modification after DAAs therapy. These results suggest complex interactions between both viruses and the immune system, which are only partially reversed by DAA treatment.

## 1. Introduction

The World Health Organization estimated that 3% of the world’s population was infected with hepatitis C virus (HCV), although the incidence of HCV on a global scale is not well known. HCV infection is prevalent among people living with HIV (PLWH), with approximately seven million people worldwide living with both infections [1]. HCV persistence in HIV/HCV coinfected subjects (HIV/HCV) is more common than in patients infected with HCV alone (mHCV), probably due to HIV-induced dysfunction of both the adaptive and innate immune system. In this setting, HCV chronic infection is characterized by a high risk of progressive damage to the liver that leads to fibrosis, cirrhosis, liver failure and finally hepatocellular carcinoma (HCC) [2,3,4].

Hepatitis C therapy has completely changed in the last decade. Until 2011, the standard of care for HCV chronic infection was based exclusively on the combination of pegylated interferon alpha (IFNα) and ribavirin (RBV) administered for 24/48 weeks with viral eradication in approximately 40–50% of treated subjects [3,4,5,6]. The success rate was even lower in HIV/HCV. The introduction in clinical practice of the interferon-free direct-acting antiviral agents (DAA) resulted in improved rates of sustained virologic response (SVR) in up to 95–100% of treated subjects, including those with HIV/HCV [7,8]. Moreover, the high rate of viral clearance among HCV monoinfected and HIV/HCV coinfected patients is independent of prior treatment experience and presence of cirrhosis [9,10]. Chronic inflammation plays a critical role in HCV-mediated liver damage [11,12]. HCV infection induces expression of inflammatory cytokines and chemokines that lead to the recruitment of inflammatory cells that infiltrate the liver, including natural killer (NK), NKT cells, regulatory T cells, monocytes/macrophages, dendritic cells (DCs) and CD4+ and CD8+ T-cells. [11,12,13,14,15,16]. Since IFN-free DAA regimens act on specific targets of the virus life cycle, without any known immunomodulatory effect, alterations in the immune milieu after treatment would likely be a direct consequence of HCV eradication. 

The aim of this study was to analyze the longitudinal modifications in cellular and soluble inflammatory markers in a cohort of mHCV and HCV/HIV coinfected patients treated with DAA compared to healthy controls (HD). We compared mHCV with HIV/HCV to evaluate if residual immune activation/depletion in HIV/HCV patients could have an impact on the effects of DAA treatment and the immunological response after eradication of HCV compared to patients without HIV infection. We assessed the following immunological parameters: activated CD4+ and CD8+ T cell rate, subsets of monocytes and dendritic cells, interferon-γ-inducible protein-10 (IP-10), soluble (s) CD163 and sCD14 plasma levels. In addition, we analyzed the plasma levels of matrix metalloproteinase (MMP)-2, which plays a key role during the inflammatory processes in viral infection and has been implicated in liver fibrogenesis. 

We found that the effects of DAAs were similar in both groups, despite differences in the immunological status and stage of fibrosis at baseline, and that HCV eradication led to substantial immune recovery in both groups.

## 2. Results

### 2.1. Demographic, Clinical and Laboratory Findings

The characteristics of the study population at baseline are shown in Table 1. The population enrolled in the study included 31 subjects with HCV-related chronic liver disease (15 with HCV mono-infection and 16 with HIV/HCV co-infection). The two groups did not differ by sex, age, HCV-RNA, lifestyle factors and platelets. As shown in Table 1, the only difference was the higher rate of liver fibrosis in the HCV group, detected both by FibroScan (kPa) and FIB-4 (*p* = 0.0114 and *p* = 0.0015, respectively). Therefore, to compare the two groups of patients, a sensitivity analysis on F4 patients of both groups was also performed for each parameter studied. Fifteen age- and sex-matched HD were included as a control group. The median age was 54 (28–68), and the ratio between males and females was 10/5. 

All subjects were treated with DAAs, with or without RBV, as shown in Table 2. Of note, all patients had an undetectable HCV viral load at SVR12.

### 2.2. Cross-Sectional Analysis 

All medians and ranges of cellular and soluble biomarkers are listed in Table 3.

#### 2.2.1. Soluble Markers of Inflammation

Plasma levels of IP-10 and sCD163 at baseline were significantly increased in HIV/HCV coinfected patients (*p* < 0.001 and *p* < 0.001, respectively) and in mHCV patients (*p* < 0.0001 and *p* < 0.0001, respectively) in comparison with HD. Conversely, levels of sCD14 were elevated only in HIV/HCV patients compared to HD (*p* < 0.05) (Figure 1). 

At baseline, plasma levels of MMP-2 were significantly increased in HIV/HCV (*p* < 0.05) and mHCV (*p* < 0.001) patients compared to HD, with significantly higher levels in mHCV than in HIV/HCV coinfected patients (*p* < 0.0009) (Figure 1). 

In the sensitivity analysis, performed on patients with advanced fibrosis (F4), we found the same differences in IP-10 and sCD163 between groups of patients, while there was no difference between HIV/HCV patients and HD in sCD14 and MMP-2 levels and between HIV/HCV and mHCV patients in MMP-2 levels (Figure 1).

#### 2.2.2. Activated T-Cells

Activation levels of CD4+ T-cells in HIV/HCV coinfected patients were significantly increased at baseline compared to mHCV patients and controls (*p* < 0.001 and *p* < 0.0001, respectively). In both groups of patients, we found higher levels of CD8+ HLA/DR+ cells at baseline compared to HD (*p* < 0.001 and *p* < 0.0001, respectively), without difference between mHCV and HIV/HCV coinfected patients. In the sensitivity analysis, there were similar differences in activation of CD4+ and CD8+ T-cells between HIV/HCV patients and the control (*p* < 0,01 for CD4+ cells and *p* < 0.001 for CD8+ cells) and in activation of CD8+ cells between mHCV patients and the control (*p* < 0.001), while there was no difference between HIV/HCV and mHCV patients (Figure 2).

#### 2.2.3. Monocyte Subsets

Intermediate monocytes were higher in mHCV patients and lower in coinfected HIV/HCV patients compared to controls (*p* = 0.058). 

When considering only patients with an advanced degree of fibrosis, intermediate monocytes were significantly higher in mHCV compared to HIV/HCV coinfected patients (*p* < 0.05, Figure 3).

#### 2.2.4. Subpopulations of DCs

In subjects with HIV/HCV co-infection, mDCs were significantly lower compared to mHCV patients and HD (*p* < 0.05 and *p* < 0.01, respectively) (Figure 4). The sensitivity analysis on subjects with an advanced degree of fibrosis showed that mDC levels were lower only in HIV/HCV coinfected patients compared to controls (*p* < 0.001).

No difference in MDC8+ and pDC levels between all groups was observed (*p* = 0.19 and *p* = 0.77, respectively, with ANOVA test). MDC8 was reduced in both groups of F4 patients compared to HD (*p* = 0.053, Figure 4).

### 2.3. Longitudinal Changes during DAA Treatment

All medians and ranges of cellular and soluble biomarkers in HIV/HCV and mHCV patients at baseline and sustained virologic response (SVR12) are listed in Table 4**.**

#### 2.3.1. Soluble Plasma Biomarkers of Inflammation and Immune Activation

In the group of mHCV patients, IP-10 and sCD163 levels decreased after treatment (*p* = 0.0012 and *p* = 0.0465, respectively), while no difference in sCD14 levels was observed (Figure 5a–c). A significant decrease in IP-10, sCD163 and sCD14 concentrations was found at SVR12 in patients with HIV/HCV infection (*p* = 0.0353, *p* < 0.0001 and *p* = 0.0248, respectively) (Figure 5d–e). To assess the degree of normalization of the soluble biomarkers at SVR12, we compared the levels of IP-10, sCD163 and sCD14 at SVR12 with HD. Despite a substantial decrease, IP-10 levels remained significantly higher in HIV/HCV patients compared to HD (*p* = 0.0008), while a normalization was observed in mHCV subjects. Differently, sCD163 levels persistently elevated in comparison with HD in treated mHCV patients (*p* = 0.0005), whereas they reached similar levels to HD in HIV/HCV subjects after treatment. Finally, sCD14 returned to similar levels to HD in coinfected subjects after treatment with DAA. No differences were found in mHCV patients. 

We also analyzed the longitudinal changes of levels of MMP-2 after DAA therapy. As shown in Figure 5d, both in the group of HCV monoinfected and HIV/HCV coinfected patients, MMP-2 plasma levels persistently elevated at SVR12 in comparison to HD (*p* < 0.0001, *p* = 0.002, respectively), with no statistically significant differences between T0 and SVR12. 

#### 2.3.2. Activated CD4+ and CD8+ T Cells

At the end of the treatment, we did not observe significant differences in the percentage of HLA-DR + CD38+ CD4+ (Figure 2a) and CD8+ T-cells (Figure 6b) in mHCV patients, while a significant reduction in activation of CD8+ T-cells was found in HIV/HCV patients (*p* = 0.0137) (Figure 6d). 

After DAA treatment, activated CD8+ T-cells were still higher in mHCV patients compared with HD (*p* = 0.0236), (Figure 6b), and activation levels of CD4+ were still higher in HIV/HCV coinfected patients compared to HD (*p* = 0.0138), (Figure 6c). Interestingly, in patients with HIV/HCV and advanced fibrosis, the rate of activation of CD8+ T-cells was similar to the controls at the end of therapy.

#### 2.3.3. Monocyte Subsets 

In mHCV patients, a significant reduction in intermediate monocytes was observed at SVR12 (*p* = 0.0122) (Figure 7b), while in HIV/HCV coinfected patients no significant differences were observed in monocyte subpopulations compared to baseline (Figure 7d–f). Interestingly, this difference in mHCV was not observed in the sensitivity analysis considering only advanced degree of fibrosis.

Compared to HD, no differences were found with mHCV at SVR12, while in subjects with HIV/HCV co-infection both intermediate and non-classical monocytes were lower at the end of treatment compared to healthy subjects (*p* = 0.003 and *p* = 0.0213, respectively) (Figure 7e,f).

#### 2.3.4. DC Subsets 

After DAAs, no statistically significant differences were observed in DCs subpopulations between the two treatment groups. Similar results were obtained in the sensitivity analysis, even if an increase in pDC and mDC was observed in HIV/HCV subjects (Figure 8). 

To evaluate the degree of normalization, we compared counts of monocytes and DCs at SVR12 with HD. In both groups, no statistically significant differences were found in all DC subpopulations compared to HD. The analysis of MDC8 cells after HCV eradication in mHCV and in HIV/HCV coinfected patients showed persistently lower levels compared with HD (Figure 8a,d).

### 2.4. Correlations between Soluble Biomarkers, Clinical Parameters and Peripheral Blood Cell Counts

Correlation analysis between biomarkers and clinical parameters was performed in all patients at baseline (Figure 9). 

We found a positive correlation between IP-10 and HCV-RNA in the total population analyzed (r = 0.5, *p* = 0.01). 

Interestingly, sCD163 positively correlated with FIB-4 and kPa (r = 0.57, *p* = 0.0006; r = 0.54, *p* = 0.002, respectively), and MMP-2 plasma levels with kPa, FIB-4, sCD163, AST and ALT (r = 0.464, *p* = 0.009681; r = 0.526, *p* = 0.0023; r = 0.67, *p* = 0.0003; r = 0.58, *p* = 0.0005; r = 0.38; *p* = 0.03) suggesting a link between these markers and liver involvement. On the other hand, sCD14 inversely correlated with AST and ALT (r = −0.52, *p* = 0.006 and r = −0.5, *p* = 0.008, respectively).

Moreover, levels of activated CD4+ inversely correlated with FIB-4, kPa, AST and sCD163 (r = −0.4, *p* = 0.02; r = −0.4, *p* = 0.02; r = −0.5, *p* = 0.007 and r = −0.5, *p*= 0.009, respectively) suggesting that CD4 T-cell activation could be protective for liver damage. 

## 3. Discussion

In the present study, we analyzed different cellular and soluble markers of inflammation and immune activation, before and after-DAA therapy, in HIV/HCV and mHCV subjects and described several changes of these markers after HCV eradication. 

The immunopathogenesis of HCV infection is characterized by an inefficient immune response that fails to eradicate the infection, with the establishment of a host-immune system balance that slowly causes liver damage. The virus-specific immune response is regulated by the innate immune system, which plays a key role in determining the establishment of chronic HCV infection and the associated hepatic fibrotic damage. Cells of the myeloid lineage, such as monocytes/macrophages, Kupffer cells and mDCs, are activated by HCV. The cells of the myeloid lineage are recruited by chemotactic factors and directly by the virus (by binding with CD81) and migrate inside the liver supporting chronic inflammation and the activation of fibrogenesis [17,18,19]. 

In order to confirm the activation status not only at the peripheral level but also in the liver, we evaluated sCD163, released from the cell surface of both monocytes/macrophages and hepatocytes following proteolytic cleavage after pro-inflammatory stimulation [20]. sCD163 is an important marker of macrophage activation [21], which is also involved in the activation of hepatic stellate cells and promotion of fibrogenesis [11,22,23,24], thus representing an ideal marker to assess both systemic and liver inflammation.

In the present study, we found significantly higher plasma levels of sCD163 in both groups of HCV and HIV/HCV patients compared to HD, confirming that activation of monocytes plays an important role in the pathogenesis of HCV infection [25]. In previous studies [26,27], high levels of sCD163 were found in subjects with HIV/HCV co-infection compared to PLWH and HD, suggesting that in the context of successful ART, immune activation could be sustained by the presence of another chronic infection such as HCV. Interestingly, sCD163 levels correlated with FIB-4 and kPa in the total population of our study, suggesting that sCD163 may represent a surrogate marker of liver fibrosis during HCV infection [28]. Of note, recent studies showed that sCD163 was higher in advanced HCV-related liver disease [29].

The activation of monocytes/macrophages was also confirmed by the increase of intermediate Mo in the two groups of mHCV and HIV/HCV patients. Intermediate monocytes represent a transitory stage of monocyte differentiation and are regarded as the mature and pro-inflammatory subset of monocytes [30,31]. Increased numbers of intermediate monocytes were observed during viral infections such as HIV and HCV and autoimmune diseases. In the context of HCV, they seem to contribute to the perpetuation of intrahepatic inflammation and profibrogenic HSC activation in liver cirrhosis [32]. The activation of monocytes/macrophages was also confirmed by the increase in sCD14 levels in HIV/HCV patients. The CD14 receptor is expressed by monocytes/macrophages, DCs and polymorphonuclear leukocytes and is secreted in a soluble form upon monocyte activation. sCD14 is a marker of microbial translocation (MT) and can be detected at high levels in patients with HCV, HIV and HIV/HCV infections [33,34]. As already demonstrated by other authors, in the early stages of HIV infection, the gastrointestinal tract mucosa can be damaged, leading to MT with subsequent systemic immune activation [35]. A decrease in gut translocation markers was observed after successful treatment with ART, even if normalization is not frequent. A relationship between microbial translocation and progression of liver disease in HCV infection has also been suggested [34,36]. Our data showed a persistent increase of sCD14 in HIV/HCV ART-treated patients that was reverted after DAA treatment, suggesting that HCV infection is the cause of persistent MT in this cohort of PLWH. 

The introduction of DAAs has resulted in improved outcomes in the management of chronic HCV infection in PLWH. Multiple clinical trials have demonstrated comparable rates of SVR12 in HIV/HCV coinfected and mHCV individuals [8,9,10,37,38,39,40,41]. The immunological response after eradication of HCV using IFN-free DAA therapies is not well defined. Several studies suggest a partial immunological recovery after treatment [42,43,44]. In our study, we found a different immune profile in the two populations in the monocyte and lymphocyte compartments as well as in levels of soluble markers after DAA treatment. 

In subjects with HIV/HCV infection, we found a significantly lower count of intermediate and non-classical monocytes in comparison with HD and a decrease in sCD163 and sCD14 levels at the end of therapy, indicating reduction in myeloid activation after HCV eradication, regardless of HIV persistence. Parisi et al. showed a substantial decrease in levels of macrophage activation markers in a cohort of DAA-treated HIV/HCV subjects regardless of the presence of low-level viremia (LLV) [45], suggesting a major role for HCV in driving immune activation in HIV/HCV patients. 

Interestingly, in mHCV patients, intermediate monocytes and sCD163 levels were also decreased after therapy, but sCD163 persisted at higher levels in comparison with HD. These findings might be related to the higher prevalence of advanced fibrosis in our population of mHCV subjects. In patients with HCV mono-infection and advanced fibrosis, we did not observe a reduction in intermediate monocytes at the end of therapy, underlining the importance of an earlier initiation of DAA treatment in an attempt to reduce liver inflammation before the establishment of advanced fibrosis. Dendritic cells (DCs) are key components of the innate immune system, able to initiate the immune response through the presentation of viral antigens to effector cells. Three major subsets of DCs can be detected in human peripheral blood: plasmacytoid (*p*)DCs, myeloid (m)DCs and MDC-8 cells, which have overlapping features with nonclassical monocytes [46,47,48]. In our study, we found a slightly lower level of circulating DCs, especially of mDC, in HIV/HCV patients at baseline, indicating persistent immune deficiency in these patients despite effective ART. At SVR12, the deficit was reverted, reaching values comparable with HD, suggesting a role of DAA treatment in the restoration of mDCs. 

We also evaluated T-cell activation and plasma levels of IP-10. IP-10 is a CXC chemokine that binds to the CXCR3 receptor on several cell types, including hepatocytes, monocytes, NK cells and T-cells, and triggers both hepatic and systemic immune activation [46]. 

It was suggested that elevated IP-10 could recruit T-cells from peripheral blood to the liver, sustaining the damage seen in advanced fibrosis [49]. In this study, in coinfected HIV/HCV subjects, the levels of activated CD4+ and CD8+ T-lymphocytes and circulating IP-10 were higher at baseline and normalized only in part after treatment with DAA. While levels of activated CD8+ T-cells decreased after treatment, levels of activated CD4+ T-cells did not change substantially. Persistent T-cell activation in HIV/HCV patients might be influenced by the presence of HIV infection [50]. On the other hand, mHCV patients had similar activated CD4+ T-cell levels to HD even at baseline, whereas IP-10 levels were elevated and decreased to levels similar to HD after DAA treatment. Activation of CD8+ T-cells was elevated at baseline and continued to be higher in comparison with HD, suggesting a persistent activation after therapy.

We also analyzed circulating levels of MMP-2, which is known to degrade the components of the extracellular matrix and to be involved in tissue remodeling and in the development of liver fibrosis during HCV infection [51,52]. In both mHCV and HIV/HCV subjects, MMP-2 plasma levels were significantly higher than HD at baseline and did not decrease significantly at SVR12, suggesting that MMP-2 levels might not be affected by the presence of HCV. This result is in line with the finding shown in a previous study from our group, where we demonstrated that MMP-2 plasma levels remained elevated after 12 months of therapy in HCV patients [52]. This result, together with the correlation between MMP-2 levels and kPa, FIB-4, aminotransferases and sCD163, suggests that MMP-2 may be used as a marker of established hepatic fibrosis, both in mHCV and HIV/HCV patients. On the other hand, the persistent high levels of some factors such as MMP-2 and sCD163 at SVR12 might also be explained by the ongoing resolution processes involved in the immune activation and chronic inflammation. 

Since all mHCV patients had a higher rate of fibrosis in comparison to HIV/HCV, to make the two groups comparable and to understand whether possible differences between mHCV and HIV/HCV were only due to the different distributions of the fibrosis stage, we also conducted a sensitivity analysis for each parameter studied, including only F4 patients from the two groups. Results obtained indicated that there were no substantial differences between the cross-sectional analysis and the sensitivity analysis at baseline. Accordingly, we found the same trend between the cross-sectional analysis and the sensitivity analysis at SVR12, although the significant differences were lost due to the small number of samples (data not shown). 

The main limitations of the present study are the small number of subjects and the higher proportion of advanced liver status in the mHCV group, which we tried to balance by performing a sensitivity analysis. Moreover, RBV was almost exclusively used in the mHCV group, in line with current HCV treatment guidelines.

## 4. Materials and Methods

### 4.1. Study Population

Subjects attending two out-patient clinics at the Department of Public Health and Infectious Diseases of the University of Rome “La Sapienza” in Rome and Latina were enrolled. The study population included 31 consecutive patients with active HCV infection (15 mHCV and 16 HIV/HCV coinfected subjects), matched by age and sex, who started HCV therapy based on the current Italian national guidelines for HCV treatment. Sustained virological response (SVR12) was defined as undetectable HCV-RNA in blood 12 weeks after the end of therapy. Inclusion criteria were absence of active HBV infection or decompensated liver disease, and for HIV-infected patients, to be on stable ART for more than 1 year and having undetectable HIV viral load (<20 copies/mL). Fifteen HD, age- and sex-matched with patients, were enrolled as a control group. The study was approved by the Lazio 2 Ethics Committee (approval number 62413, study number 99/16). All subjects signed a written informed consent before enrolment in the study. Data and plasma samples were collected respecting donor’s confidentiality and privacy.

### 4.2. Sample Collection

Peripheral blood samples were collected before starting DAA (T0) and 12 weeks after the end of treatment (SVR12). Venous blood samples were collected into EDTA and heparin-containing tubes (Becton ± Dickinson Systems, San Jose, CA, USA), and the cell-free plasma was stored at −800 °C until use. 

### 4.3. Clinical and Laboratory Assessment

Liver stiffness was evaluated by non-invasive transient elastography (Fibroscan Central, FS Echosens, Paris, France) and measured in kPa. Advanced fibrosis (F4) was defined as a score of liver stiffness greater than or equal to 14.5 kPa in HCV patients and 12.5 in HIV/HCV patients; severe fibrosis (F3) was defined as a score between 9.5 and 14.5 kPa in HCV patients and 11 and 12.5 in HIV/HCV patients; mild to absent fibrosis (F0–F2) was defined as a score on liver stiffness of less than 9.5 kPa in HCV patients and 11 kPa in HIV/HCV patients [53]. Plasma HCV-RNA levels were determined by Real-Time PCR Roche Cobas TaqMan. HCV genotypes were determined by Abbott Real-Time HCV Genotype II. To assess hepatic fibrosis, we also used a biochemical index (FIB-4), calculated using Sterling’s formula: age [years] × AST [IU/L]/platelet count [expressed as platelets × 109/L] × (ALT 1/2 [IU/L]). In all patients, we assessed serum levels of aspartate aminotransferase (AST), alanine aminotransferase (ALT), platelets (PLT), albumin, cholesterol, creatinine, bilirubin and international normalized ratio (INR). PLWH were tested for HIV-RNA, CD4+ and CD8+ T-cell counts. We also collected information on smoking habits, alcohol use and body mass index (BMI) calculated as weight (kilograms) divided by height (meters) squared. Subjects were considered as high-risk alcohol consumers if they reported intake of 5 or more alcoholic beverages on a single occasion in the previous 30 days. Dyslipidemia was defined based on medical records and/or presence of fasting total cholesterol > 200 mg/dL and/or fasting triglycerides >150 mg/dL. 

### 4.4. CD4+ and CD8+ T-Cells Activation Markers 

Immune activation of peripheral blood CD4+ and CD8+ T lymphocytes was evaluated by direct staining of whole blood. A volume of 50 μL of blood was incubated with fluorochrome-conjugated monoclonal antibodies for 25 min at room temperature (RT) in the dark. Lysing solution (BD Biosciences Pharmagen, Milano, Italy) was added to the samples, and after additional 10 min of incubation, the samples were analyzed using a MACS Quant flow cytometer (Miltenyi Biotec, Bergisch Gladbach, Cologne, Germany). The anti-human monoclonal antibodies used were CD45-VioBlue, HLA-DR-FITC, CD38-APC, CD8-PerCP and CD4-PE (all from Miltenyi Biotec, Bergisch Gladbach, Germany). SSC and CD45 were used to identify lymphocytes. CD4+ and CD8+ T-cells were identified from lymphocytes, and finally, activated CD4+ and CD8+ T-cells were identified by the expression of both HLA-DR and CD38.

### 4.5. Flow Cytometry Analysis of Peripheral Blood Monocytes and Dendritic Cells 

Peripheral blood monocyte and DCs subsets were analyzed using flow cytometry, as previously described [54]. To identify the number of cells, a lyse no-wash protocol was used. A volume of 50 μL of whole blood was stained with the following monoclonal antibodies: CD3-PerCP (clone BW264/56), CD20-PerCP (clone LT20), CD56-PerCP (clone REA196), HLA-DR-APC (clone AC122) and CD14-VioGreen (clone TUK4) from Miltenyi Biotec, Germany; CD235a-PerCP (clone HI264, diluted 1:100 with phosphate buffer saline), CD16-PE Vio770 (clone 3G8) and CD123-BV from BioLegend, Inc. San Diego, USA; CD11c-PE (CLONE) from BD Biosciences Pharmagen, Italy. After 30 min of incubation, at RT and in the dark, lysing solution was added and, after 10 min of incubation, the samples were analyzed using a MACS Quant flow cytometer. To identify monocyte subsets, we used the following gating strategy: lymphomonocytes (R1 gate) were gated in a dot plot of side scatter channel (SSC) versus forward scatter channel (FSC) dot plot; then, FSC-A and FSC-H were selected to obtain single cells (R2). The lineage cocktail (CD3, CD20 and CD235a) was used to exclude B and T-lymphocytes, NK cells and erythrocytes (R3); from gate R3, monocytes were defined as HLA-DR+. Monocyte surface markers CD14 and CD16 were used to identify the three monocyte subsets: classical monocytes (CM) defined as CD14+/CD16−; intermediate monocytes (IM) defined as CD14+/CD16+; non-classical monocytes (NCM) defined as CD14low/CD16+. DCs markers CD123 and CD11c were used as negative cells for both CD14 and CD16 to identify the plasmacytoid dendritic cells (pDC) and myeloid dendritic cells (mDC), respectively; MDC8 and CD11c were used to identify slanDCs from CD14lowCD16+ cells. 

### 4.6. Soluble Markers of Inflammation and Plasma Matrix Metalloproteinases 

Interferon-inducible protein-10 (IP-10), sCD163 and sCD14 were measured by ELISA (Quantikine human, R&D Systems, Minneapolis, MN), as previously described [26,55]. The reported minimum detectable levels of IP-10, sCD163 and sCD14 were 1.67 pg/mL, 0.177 ng/mL and 0.125 ng/mL, respectively. All samples were tested in duplicate.

Plasma levels of MMP-2 were detected by zymography as described by Iannetta et al. (2019) [56]. Briefly, 1 µL of plasma sample was dissolved in 10 µL of electrophoresis loading buffer containing sodium dodecyl sulfate (SDS) and then separated in 10% polyacrylamide slab gels copolymerized with 0.1% (*w*/*v*) type A gelatin from porcine skin (Sigma Chemical Co., St. Louis, MO, USA). Stacking gels contained 5.4% polyacrylamide. The gels were run at 4 °C for 2 h at 100 V. After electrophoresis, the gels were washed for 2 × 30 min in 2.5% (*w*/*v*) Triton X-100/10 mM CaCl2 in 50 mM Tris -HCl, pH 7.4 (washing buffer), in order to remove SDS, and then incubated for 24 h at 37 °C in 1% (*w*/*v*) Triton X-100/50 mM Tris -HCl/10 mM CaCl2, pH 7.4 (developing buffer). After staining and de-staining of gels, MMP-2 and MMP-9 plasma levels were detected as white bands of digestion on the blue background of the gel and were identified by co-localization with MMP-2 standard (ALEXIS Biochemicals, San Diego, CA, USA). Quantification of MMP-2 levels was performed using computerized image analysis (Image Master 1D, Pharmacia Biotech, Buckinghamshire, UK) through one-dimensional scanning densitometry (Ultroscan XL, Pharmacia Biotech). MMP-2 levels were expressed as optical density (OD) × mm^2^, representing the scanning area under the curves, which considers both brightness and width of the substrate lysis zone. 

### 4.7. Data Analyses and Statistics 

Flow Cytometry data were analyzed using FlowJo Software (FlowJo v 9.2, Tree Star, Ashland, Oregon, USA). All statistical analyses were performed using GraphPad Prism version 6.0 for Windows (GraphPad Software MacKiev). Values are described as median and range (minimum and maximum value). The 2-tailed χ2 test or Fisher’s exact test was used to compare binominal variables. The Mann–Whitney test and non-parametric Kruskal–Wallis ANOVA with Dunn’s post-test were used to compare differences in values between two or more groups. Non-parametric Wilcoxon matched pairs test was used to perform longitudinal analyses. Spearman’s rank correlation coefficient (r) was calculated to determine associations between variables. Differences were considered significant if *p* ≤ 0.05.

## 5. Conclusions

In the present study, we described a different immune activation signature and several changes in T-cell and monocyte-mediated immunity and in soluble factors of inflammation after HCV eradication obtained by DAA, in HIV/HCV coinfected and mHCV subjects. In mHCV patients, we found some alterations that also persisted after therapy such as an up-regulation of activated CD8+ T cells, sCD163 and MMP-2, whereas intermediate monocytes and IP-10, that were increased at baseline, normalized after DAA treatment. In HIV/HCV subjects on effective ART, after HCV eradication, we observed down-regulation of activated CD8+ T cells, sCD163 and sCD14 plasma level and increase of pDC and mDC, suggesting a pivotal role of HCV replication in the persistent immune activation despite HIV control. In contrast, activated CD4+ T cells, IP-10 and MMP-2 seemed to be unchanged by treatment and persistently elevated in comparison with HD. All these results suggest that the immunological differences that persist despite the eradication of HCV in both groups could be due to the advanced fibrotic status in mHCV and to the ongoing HIV replication despite ART in HIV/HCV coinfected patients. The evidence of persisting immune activation after HCV eradication both in mHCV and HIV/HCV patients with different patterns could have important clinical implications in terms of monitoring of immune-mediated end-organ diseases associated with HCV and HIV infections. A longer follow-up and a larger cohort of patients might help to clarify whether inflammation resolves over time or persists after HCV eradication.

## Figures and Tables

**Figure 1 ijms-23-11936-f001:**
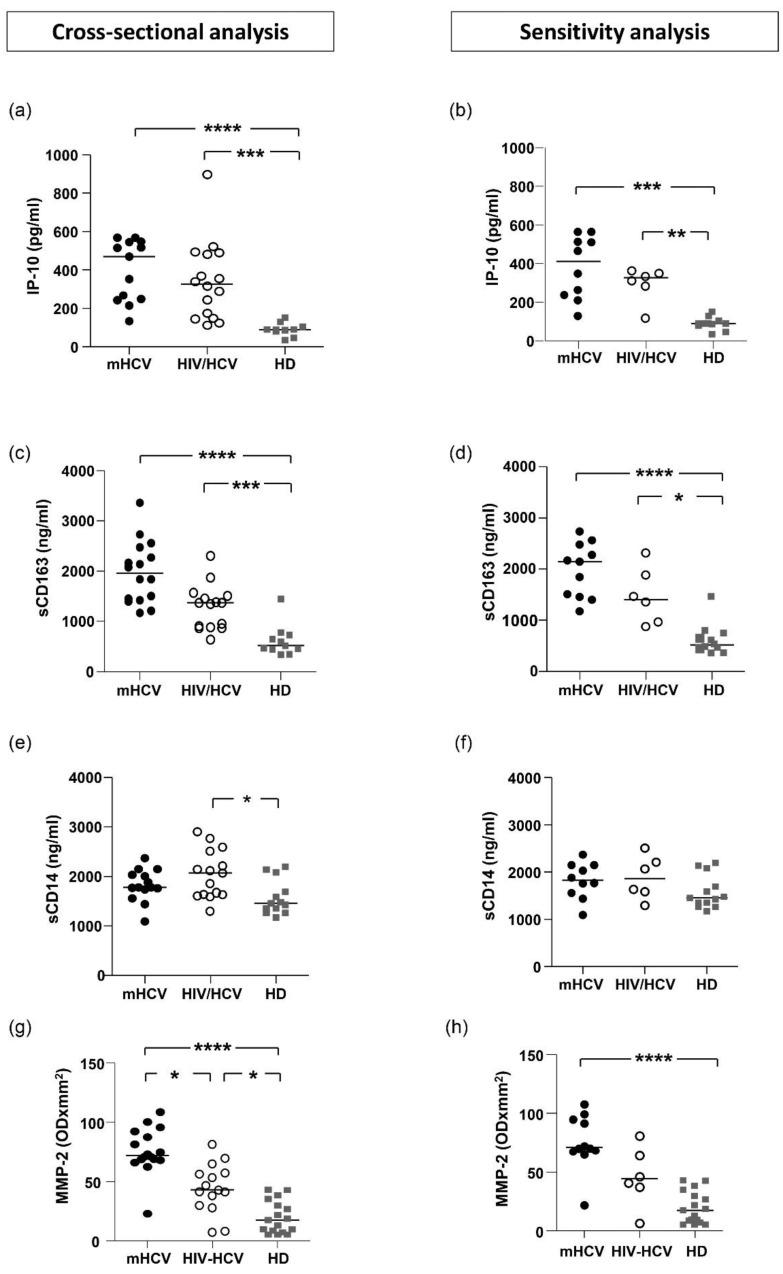
**Plasma levels of IP-10, sCD163, sCD14 and MMP-2 in mHCV and HIV/HCV patients, at baseline, compared to healthy controls.** Box plots show circulating levels of IP-10 (**a**,**b**), sCD163 (**c**,**d**), sCD14 (**e**,**f**) and MMP-2 (**g**,**h**), at baseline, in the total population of mHCV and in HIV/HCV patients (**a**,**c**,**e**,**g**) and in mHCV and HIV/HCV patients with advanced degree of fibrosis (**b**,**d**,**f**,**h**). Horizontal bars represent the median values. Kruskal–Wallis ANOVA with Dunn’s post-test was used to assess statistically significant differences between the groups. HD, healthy donors; * = *p* < 0.05; ** = *p*< 0.01; *** = *p*< 0.001; **** = *p*< 0.0001.

**Figure 2 ijms-23-11936-f002:**
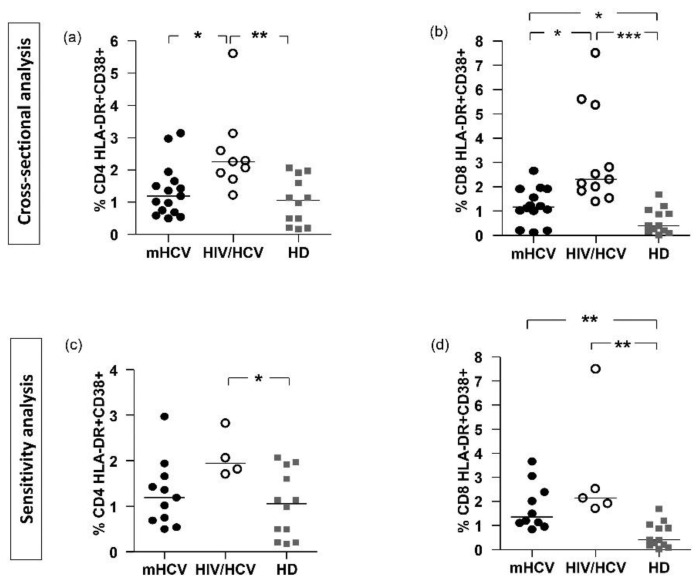
**Percentage of immune activation of CD4 + and CD8 + in mHCV and HIV/HCV coinfected patients, at baseline, compared to healthy controls.** Box plots represent circulating percentage of CD4+ (**a**,**c**) and CD8+ (**b**,**d**) activation, at baseline, in the total population of mHCV and HIV/HCV patients (**a**,**b**) and in mHCV and in HIV/HCV patients with advanced degree of fibrosis (**c**,**d**). Horizontal bars represent the median values. Kruskal–Wallis ANOVA with Dunn’s post-test was used to assess statistically significant differences between the groups. HD, healthy donors; * = *p* < 0.05; ** = *p*< 0.01; *** = *p* < 0.001.

**Figure 3 ijms-23-11936-f003:**
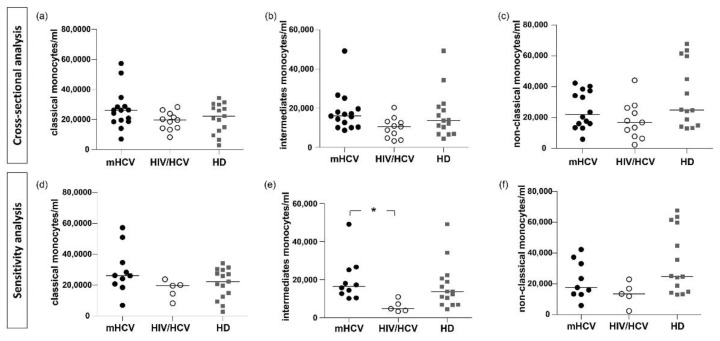
**Counts of classical, intermediate and non-classical monocytes in mHCV and HIV/HCV coinfected patients, at baseline, compared to healthy controls.** Box plots show counts of circulating classical, intermediate and non-classical monocytes, at baseline, in the total population of mHCV and in HIV/HCV patients (**a**–**c**) and in mHCV and in HIV/HCV patients with advanced degree of fibrosis (**d**–**f**). Horizontal bars represent the median values. Kruskal–Wallis ANOVA with Dunn’s post-test was used to assess statistical differences between the groups. HD, healthy donors; * = *p* < 0.05.

**Figure 4 ijms-23-11936-f004:**
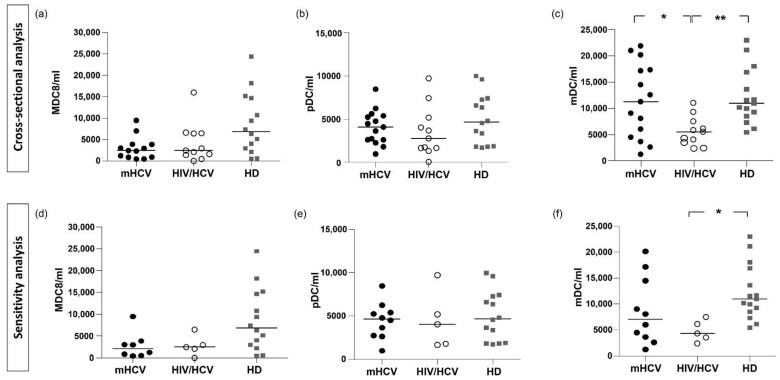
**Counts of MDC8, pDC and mDC in mHCV and HIV/HCV coinfected patients, at baseline, compared to healthy controls.** Box plots show circulating counts of MDC8, pDC and mDC, at baseline, in the total population of mHCV and HIV/HCV patients (**a**–**c**) and in mHCV and in HIVHCV patients with advanced degree of fibrosis (**d**–**f**). Horizontal bars represent the median values. Kruskal–Wallis ANOVA with Dunn’s post-test was used to assess statistical differences between the groups. HD, healthy donors; * = *p* < 0.05; ** = *p*< 0.01.

**Figure 5 ijms-23-11936-f005:**
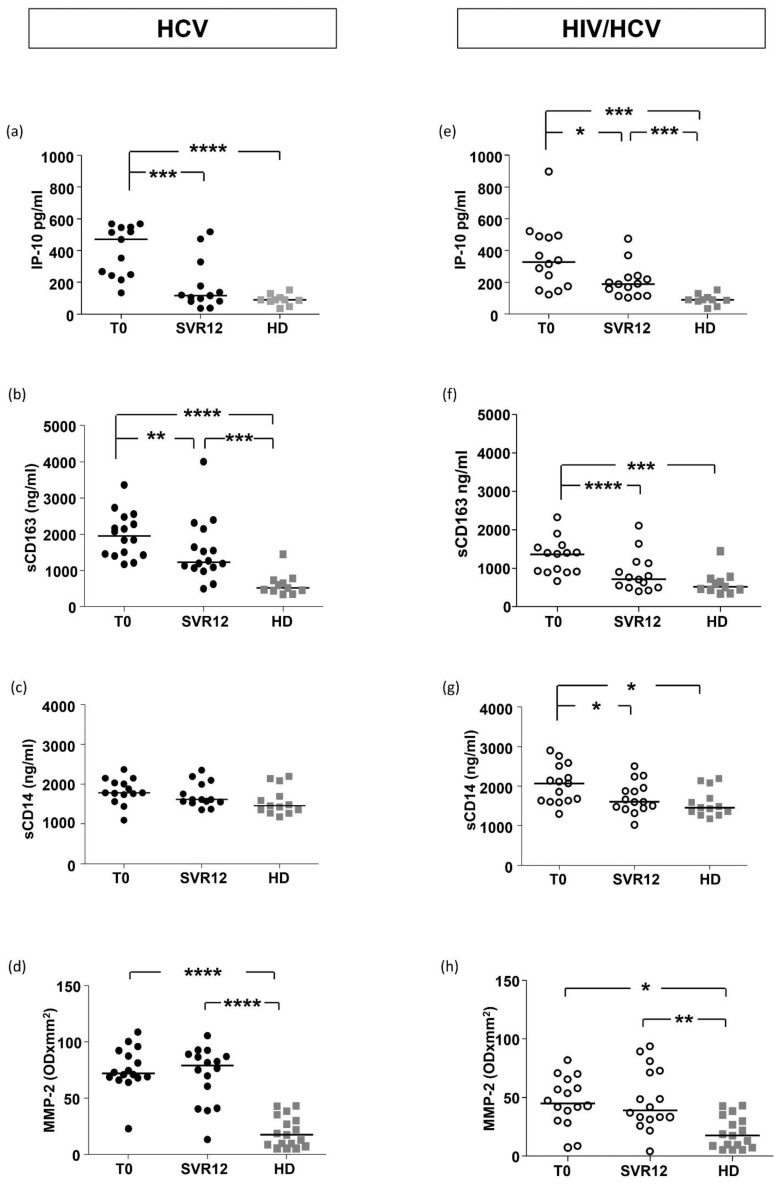
**Plasma levels of IP-10, sCD163, sCD14 and MMP-2 in mHCV patients and HIV/HCV patients, during and after anti-HCV therapy with DAAs, compared to healthy controls.** Box plots show circulating levels of IP-10, sCD163, sCD14 and MMP-2 in mHCV patients (**a**–**d**) and in HIV/HCV patients (**e**–**h**) during and after therapy with DAAs. Horizontal bars represent the median values. Wilcoxon test was performed to assess differences between the baseline and SVR12, while Mann–Whitney test assessed differences between SVR12 and HD. HD, healthy donors; T0, baseline before therapy; SVR12, sustained virologic response 12 weeks after the end of therapy. * = *p* < 0.05; ** = *p*< 0.01; *** = *p*< 0.001; **** = *p*< 0.0001.

**Figure 6 ijms-23-11936-f006:**
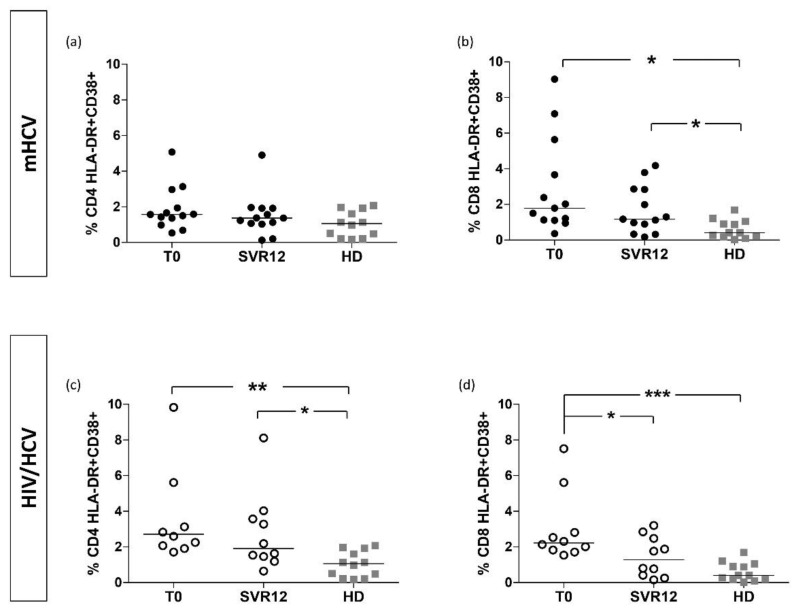
**Percentage of immune activation of CD4 + and CD8 + in mHCV patients and HIV/HCV patients during and after anti-HCV therapy with DAAs, compared to healthy controls.** Box plots represent percentage of circulating CD4+ and CD8+ lymphocytes in mHCV patients (**a**,**b**) and in HIV/HCV coinfected patients (**c**,**d**) during and after therapy with DAAs. Horizontal bars represent the median values. ANOVA with Dunn’s post-test was performed to assess differences between patients at baseline and controls, Wilcoxon test was performed to assess differences between the baseline and SVR12 and Mann–Whitney test assessed differences between SVR12 and HD. HD, healthy donors; T0, baseline before therapy; SVR12, sustained virologic response 12 weeks after the end of therapy. * = *p*<0.05; ** = *p* < 0.01; *** = *p* < 0.001.

**Figure 7 ijms-23-11936-f007:**
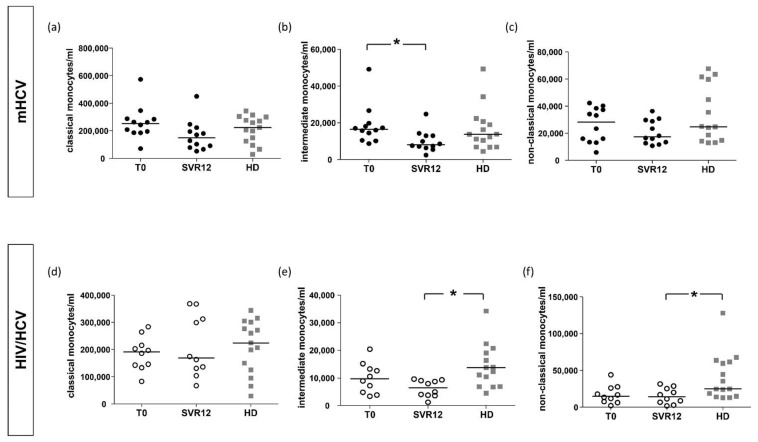
**Counts of classical, intermediate and non-classical monocytes in mHCV patients and HIV/HCV patients, during and after anti-HCV therapy with DAAs, compared to healthy controls (HD).** Box plots show counts of circulating classical, intermediate and non-classical monocytes in HCV infected patients (**a**–**c**) and in HIV/HCV coinfected patients (**d**–**f**) during and after therapy with DAAs. Horizontal bars represent the median values. ANOVA with Dunn’s post-test was performed to assess differences between patients at baseline and control, Wilcoxon test was performed to assess differences between the baseline and SVR12 and Mann–Whitney test assessed differences between SVR12 and HD. HD, healthy donors; T0, baseline before therapy; SVR12, sustained virological response 12 weeks after the end of therapy. * = *p* < 0.05.

**Figure 8 ijms-23-11936-f008:**
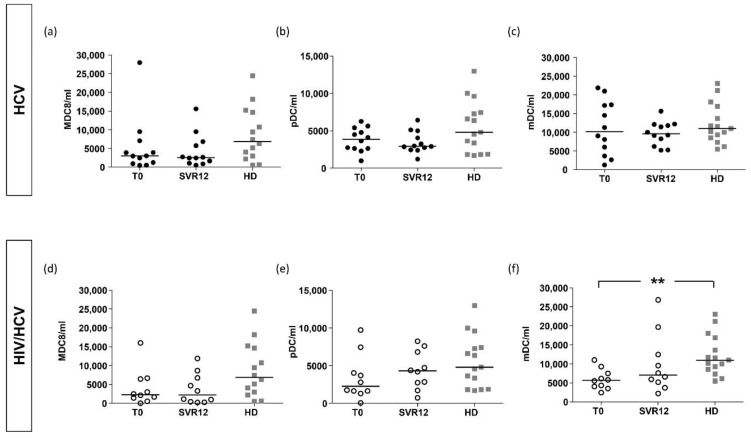
**Counts of MDC8, pDC and mDC in mHCV and HIV/HCV patients, during and after anti-HCV therapy with DAAs, compared to healthy controls.** Box plots show circulating counts of MDC8, pDC and mDC in mHCV patients (**a**–**c**) and in HIV/HCV coinfected patients (**d**–**f**) during and after therapy with DAAs. Horizontal bars represent the median values. ANOVA with Dunn’s post-test was performed to assess differences between patients at baseline and control, Wilcoxon test was performed to assess differences between the baseline and SVR12 and Mann–Whitney test assessed differences between SVR12 and HD. HD, healthy donors; T0, baseline before therapy; SVR12, sustained virologic response 12 weeks after the end of therapy. ** = *p* < 0.01.

**Figure 9 ijms-23-11936-f009:**
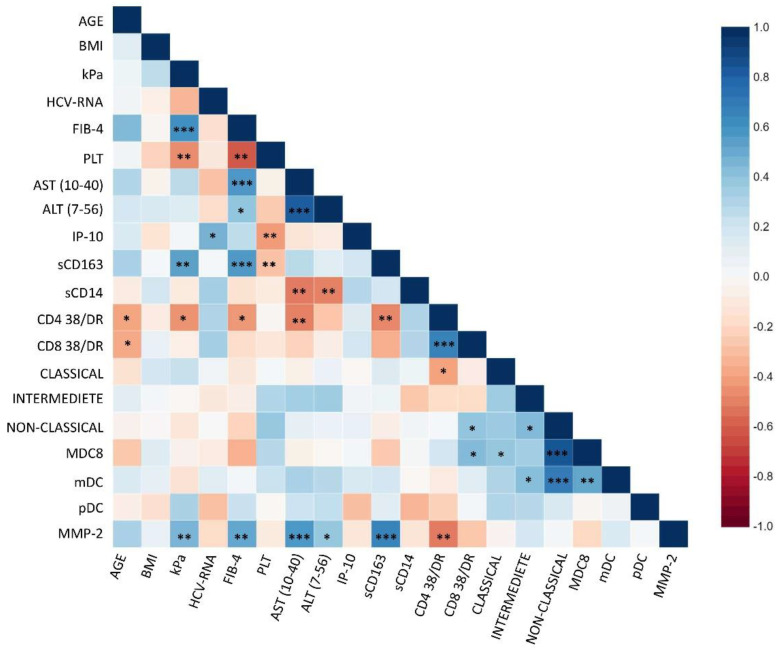
**Correlation analysis between soluble inflammatory factors, clinical parameters and peripheral blood cell counts.** Correlation analysis in the total population of mHCV and HIV/HCV coinfected patients was performed, at baseline, by the Spearman’s rank correlation coefficient (r). Colors express the strength of correlation by r, and asterisks indicate the *p* value (* = *p*<0.05; ** = *p* < 0.01; *** = *p* < 0.001).

**Table 1 ijms-23-11936-t001:** Demographic and clinical characteristics of mHCV and HIV/HCV coinfected patients at baseline.

Characteristics	mHCV (*n* = 16)	HIV/HCV (*n* = 15)	*p* Value
Age (Years)	62 (28–72)	53 (42–59)	NS
Sex, F (*n*,%)	2 (13)	6 (40)	NS
Stiffness (kPa)	16.50 (10.9–45.7)	11.10 (5.6–41.7)	0.0114
F0–F2	-	8	-
F3	5	1	-
F4	11	6	-
FIB-4	3.770 (2.199–5.736)	1.740 (1.05–6.74)	0.0015
AST level, IU/L	75 (26–170)	36 (22–103)	0.0039
ALT level, IU/L	86 (28–407)	42 (17–187)	NS
PLT (10^9^ L)	140 (81–280)	171 (57–301)	NS
HCV-RNA (UI/mL)	718,809 (57,472–5,259,000)	1,486,000 (148,100–23,110,000)	NS
HCV Genotype (*n*. %)			
-1a	4 (25)	7 (46)	NS
-1b	6 (37)	3 (20)	NS
-1a/1b	0	1 (7)	NA
-2a/2c	3 (19)	0	NA
-3a	2 (13)	4 (27)	NS
-4	1 (6)	0	NA
HIV-RNA undetectable (*n*,%)	NA	15 (100%)	NA
CD4 Nadir (mmc)	NA	102 (7–549)	NA
Actual CD4+ (mmc)	NA	670.5 (144–1736)	NA
Alcohol (*n*, %)			
Yes	5 (31%)	5 (27%)	NS
No	11 (69%)	10 (73%)	NS
Smoke (*n*, %)			NS
Yes	5 (31%)	11 (33%)	NS
No	11 (69%)	4 (67%)	
BMI (kg/m^2^)	27.3 (20.9–33.2)	25.5 (17.7–38.1)	NS

Results are expressed in median (range). For continuous variable Mann–Whitney test and for dichotomous variables, chi-square tests were performed. Abbreviations: HCV, hepatitis C virus; HIV, human immunodeficiency virus; HCV-RNA, hepatitis C virus ribonucleic acid; HIV-RNA, human immunodeficiency virus ribonucleic acid; FIB-4: fibrosis score; ALT, alanine aminotransferase; AST, aspartate aminotransferase; PLT, platelets; CD4, cluster of differentiation 4; BMI, body mass index; F0–F2, mild to absent fibrosis; F3, severe fibrosis; F4, advanced fibrosis; NA, not applicable; NS, not significant.

**Table 2 ijms-23-11936-t002:** DAA regimes used in mHCV and HCV/HVC coinfected patients.

ANTI-HCV therapy	mHCV (*n* = 16)	HCV/HIV (*n* = 15)
3D	2	8
3D + RBV	2	0
SOF + SIMV	7	1
SOF + SIM + RBV	1	0
SOF + DAC	0	1
SOF + RBV + LED	0	1
SOF + RBV	4	2
SOF + LED	0	2

SOF: sofosbuvir; SIM: simeprevir; LED: ledipasvir; 3D: ombitasvir, paritaprevir, dasabuvir; DAC: daclatasvir; RBV: ribavirin.

**Table 3 ijms-23-11936-t003:** Median values and range of immune activation of CD4 + and CD8 +, monocytes, DCs and soluble factors in mHCV and HIV/HCV patients at baseline compared to healthy donors.

Parameters	mHCV (*n* = 16)	HIV/HCV (*n* = 15)	HD	*p* Value ANOVA Test	*p* Value
IP-10 (pg/mL)	470.6 (133.8–568.5)	327.3 (111.8–897.4)	89.87 (35.76–152.3)	<0.0001	<0.001 ^b^ <0.0001 ^c^
sCD163 (ng/mL)	1957 (1169–3360)	1372 (643.3–2308)	514.6 (339–1445)	<0.0001	<0.001 ^b^ <0.0001 ^c^
sCD14 (ng/mL)	1779 (1093–2368)	2069 (1296–2901)	1455 (1172–2195)	0.0206	<0.05 ^b^
MMP-2 (ODxmm^2^)	72 (22.9–108.7)	45.05 (7.4–81.7)	17.59 (37.79–85.33)	<0.0001	<0.05 ^a^ <0.05 ^b^ <0.0001 ^c^
CD4+ CD38+ DR+ (%/mL)	1.28 (0.5–3.14)	2.18 (1.71–13.4)	1.06 (0.17–2.07)	0.0006	<0.05 ^a^ <0.01 ^b^
CD8+ CD38+ DR+ (%/mL)	1.5 (0.84–9.03)	2.23 (1.54–7.51)	0.41 (0.024–1.69)	<0.0001	<0.05 ^a^ <0.001 ^b^ <0.05 ^c^
Classical monocytes/mL	261,450 (70,091–573,055)	196,812 (827,585–283,659)	241,798 (65,642–343,801)	NS	NS
Non-classical monocytes/mL	23,345 (13,064–42,259)	16,756 (2215–44,020)	24,794 (12,965–67,569)	NS	NS
Intermediate monocytes/mL	16,018 (8747–49,132)	10,565 (3351–203,34)	13,724 (4487–49,231)	NS	NS
MDC8/mL	2954 (454–9486)	2442 (0.0–16,018)	6844 (526–24,411)	NS	NS
mDC/mL	11,246 (1250–21,868)	5510 (2386–11,019)	10962 (5453–23,010)	0.0072	<0.05 ^a^ <0.01 ^b^
pDC/mL	4090 (965.6–8463)	2783 (56.8–9713)	4672 (1704–9986)	NS	NS

Results are expressed as median (range). Statistical analyses were carried out using Kruskal–Wallis ANOVA followed by Dunn’s post-test. HD: healthy donors. NS: not significant. a: statistically significant differences between mHCV and HIV/HCV patients; b: statistically significant differences between HIV/HCV patients and HD; c: statistically significant differences between mHCV patients and HD.

**Table 4 ijms-23-11936-t004:** Median values and range of immune activation of CD4+ and CD8+, monocytes, DCs and soluble factors in HIV/HCV and HCV patients at baseline and after anti-HCV therapy.

Median (range)	mHCV		HIV/HCV		*p* Value
	T0	SVR12	T0	SVR12	
IP-10 (pg/mL)	470.6 (133.8–568.5)	116 (36–517.6)	327.3 (111.8–897.4)	189.2 (103.8–474.6)	0.0353 ^a^ 0.0012 ^b^
sCD163 (ng/mL)	1957 (1169–3360)	1228 (491.7–4002)	1372 (643.3–2308)	711.7 (375–2087)	<0.0001 ^a^ 0.00465 ^b^
sCD14 (ng/mL)	1779 (1093–2368)	1613 (1356–2348)	2069 (1296–2901)	1604 (1019–2504)	0.0248 ^a^
MMP-2 (ODxmm^2^)	72 (22.9–108.7)	78.4 (13.4–105.6)	45.05 (7,4–81.7)	39.1 (3.7–93.5)	NS
CD4+ CD38+ DR+ (%/mL)	1.275 (0.5–3.14)	1.17 (0.13–2.66)	2.175 (1.710–13.40)	1.91 (0.65–8.11)	NS
CD8+ CD38+ DR+ (%/mL)	1.5 (0.84–9.03)	1.19 (0.12–3.790)	2.225 (1.54–7.51)	1.28 (0.15–3.2)	0.0137 ^a^
Classical monocytes/mL	261,450 (70,091–573,055)	149,640 (52,597–450,026)	196,812 (827,585–283,659)	168,355 (67,138–368,291)	NS
Non-classical monocytes/mL	23,345 (13,064–42,259)	17,267 (10,678–36,238)	16,756 (2215–44,020)	14,172 (1818–31,581)	NS
Intermediate monocytes/mL	16,018 (8747–49,132)	8037 (2442–24,780)	10,565 (3351–20,334)	6447 (1193–9599)	0.0122 ^b^
MDC8/mL	2954 (454.4–9486)	2442 (511.2–9486)	2442 (0.0–16,018)	2187 (170.4–11,871)	NS
mDC/mL	11246 (1250–21,868)	9599 (5169–15,620)	5510 (2386–11,019)	7072 (2215–26,810)	NS
pDC/mL	4090 (965.6–8463)	2925 (1193–6418)	2783 (56.8–9713)	4288 (738.4–8236)	NS

Results are expressed in median (range). Statistical analyses were carried out using Wilcoxon test. NS: not significant; T0: baseline before therapy; SVR12: sustained virologic response 12 weeks after the end of therapy. a: Statistically significant difference between T0 and SVR12 in HIV/HCV; b: statistically significant difference between T0 and SVR12 in mHCV.

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
