# Peer review of "Longitudinal Assessment of Multiple Immunological and Inflammatory Parameters during Successful DAA Therapy in HCV Monoinfected and HIV/HCV Coinfected Subjects"

_ijms, 2022, doi:10.3390/ijms231911936_

Round 1

Reviewer 1 Report

In this study, authors describe changes in immune cells and soluble markers in peripheral blood of patients with HCV and HIV/HCV infections upon DAA treatment (with or without RBV). The subject is of current interest given that the host’s immune state in chronic infection along with recovery are not fully understood. With the DAAs being the first curative therapies for a chronic viral infection, studying the pre/post HCV cure states in various settings can provide a deeper understanding of the biology of the chronic immune activation and its resolution, and might open insights to comprehensive therapies and/or prevention. 

While the study is of current topic, it is unfortunate that the design is not well developed, and the observed data need more clarity and in depth interpretation.    

1) The groups as they are presented are not comparable: e.g., HCV mono-infected patients have a more advanced stage of liver disease and more of them received RBV (compared to HIV/HCV co-infected patients). Due to this start, the comparison of the data between the two groups is compromised and difficult to interpret. Furthermore, the possibly relevant data (sensitivity analysis) are placed into supplementary figures.  

- To assess results between comparable groups, the authors could compare parameters (baseline and changes) between HIV/HCV and HCV, within each of the 2 groups of liver disease stage (F0-F2) and (F3-F4). The liver disease groups could also be F0-F1 (none to mild fibrosis) and F2-F4 (advanced fibrosis, cirrhosis) 

- How many subjects are there per these liver disease stages in each group? 

- How are the liver disease stages assessed? Is it by biopsy, in both HCV and HIV/HCV? This is not mentioned in the text. 

- Is there an available histological assessment of inflammation?  

2) Authors conclude that the studied events are “only substantially reversed by DAA”. However, an evaluation of longer follow-up time-points is needed to make such a conclusion e.g., 24 and/or 48 weeks post SVR.  

- Events at SVR12 might be explained by the ongoing resolution processes which involve macrophages, MMPs etc. and can offer an explanation for the persistent high levels of MMP-2 or abnormal levels of sCD163 at SVR12.  

3) What is the point of showing the CD4 Nadir in HIV patients? Does it correlate with any parameters in this study?

Author Response

REVIEWER #1

Comment:

While the study is of current topic, it is unfortunate that the design is not well developed, and the observed data need more clarity and in depth interpretation.

1) The groups as they are presented are not comparable: e.g.,HCV mono-infected patients have a more advanced stage of liver disease and more of them received RBV (compared to HIV/HCV co-infected patients). Due to this start, the comparison of the data between the two groups is compromised and difficult to interpret. Furthermore, the possibly relevant data (sensitivity analysis) are placed into supplementary figures.

- To assess results between comparable groups, the authors could compare parameters (baseline and changes) between HIV/HCV and HCV, within each of the 2 groups of liver disease stage (F0-F2) and (F3-F4). The liver disease groups could also be F0-F1 (none to mild fibrosis) and F2-F4 (advanced fibrosis, cirrhosis)

Response:

The point that the Reviewer 1 underlines is a fundamental one to understand the study.

As reported in Session 4.1 “Study population” the study design was a longitudinal and prospective study that enrolled all consecutive mHCV and HIV/HCV infected patients starting DAA, only matched by age and sex (lines 475-477).

As shown in Table 1 all mHCV patients had a F3-F4 fibrosis rate. Therefore, to make the two groups comparable, even numerically, and to understand if the differences between mHCV and HIV/HCV were only due to the different distribution of the fibrosis stage, we have conducted also s sensitivity analysis for each parameter studied, including only F4 patients from the two groups. Results obtained indicated that there were no substantial differences between the cross-sectional analysis and the sensitivity analysis at baseline. These results are shown in the first version of the manuscript in Figures 1S-4S, that, according to the referee suggestion, have been now included in the main manuscript (Figures 1-4). Accordingly, we found the same trend between the cross-sectional analysis and the sensitivity analysis at SVR12, although the significant differences were lost due to the small number of samples. We added a sentence in the discussion to explain this result (lines 456-464).

The Reviewer also suggested to compare parameters (baseline and changes) between HIV/HCV and HCV, within each of the 2 groups of liver disease stage (F0-F2) and (F3-F4).   Unfortunately, we could not compare patients with F0-F2 stage between the two groups because there weren’t any F0-F2 patients in the mHCV group.

Regarding the effect of RBV, this drug was used mostly in mHCV, as shown in the figure below, in line with the guidelines at the time of the study, so we could not control for this factor in the analysis. We added this point as a limitation in the manuscript (lines 467-468 at the end of the discussion). The figure below reports the non-DAA therapies in both groups.

Comment:

- How many subjects are there per these liver disease stages in each group?

Response:

All the patients in the mHCV group were in stage F3-F4, while in HIV/HCV only 7patients were F3-4 and 8 F0-F2. These data have been now added in Table 1. The exact distribution of patients according to the degree of fibrosis is reported in the figure below.

Comment:

- How are the liver disease stages assessed? Is it by biopsy, in both HCV and HIV/HCV? This is not mentioned in the text.

Response:

In the section 4.3 “Clinical and laboratory assessment” of Material and Methods we specified that liver stiffness was evaluated by non-invasive transient elastography (Fibroscan Central, FS Echosens, France) and measured in kPa (lines 495-501). As reported, advanced fibrosis (F4) was defined as a score of liver stiffness greater than or equal to 14.5 kPa in HCV patients and 12.5 kPa in HIV/HCV patients; severe fibrosis (F3) was defined as a score between 9.5-14.5 kPa in HCV patients and 11-12.5 kPa in HIV/HCV patients; mild to absent fibrosis (F0-F2) was defined as a score of less than 9.5 kPa in HCV patients and 11 kPa in HIV/HCV patients. Liver biopsy was not performed in this cohort of patients in line with National and International Guidelines (Khalifa A, Rockey DC.“The utility of liver biopsy in 2020. Curr Opin Gastroenterol 2020;36:184–91” and Neuberger J, et al. Gut 2020;69:1382–1403. doi:10.1136/gutjnl-2020-321299.)

Comment:

- Is there an available histological assessment of inflammation?

Response:

Liver biopsies were not performed as part of the study, therefore no information on histological characteristics of inflammation was available.

Comment:

2) Authors conclude that the studied events are “only substantially reversed by DAA”. However, an evaluation of longer follow-up time-points is needed to make such a

conclusion e.g., 24 and/or 48 weeks post SVR.

Response:

We agree with the reviewer that a longer follow-up would be needed, and this was specified in the conclusions: “A longer follow-up and a larger cohort of patients could be necessary to clarify whether inflammation will resolve over time or persist after HCV eradication.” (lines 599-601).

Comment:

- Events at SVR12 might be explained by the ongoing resolution processes which involve macrophages, MMPs etc. and can offer an explanation for the

persistent high levels of MMP-2 or abnormal levels of sCD163 at SVR12.

Response:

Following the suggestion of the reviewer we have added a sentence to better explain that the persistent high levels of some parameters such as MMP-2 and sCD163 might be explained by the ongoing resolution processes involved in the immune activation and chronic inflammation (lines 453-455).

Comment:

3) What is the point of showing the CD4 Nadir in HIV patients? Does it correlate with any parameters in this study?

Response:

We showed CD4 nadir in HIV patients in Table 1 because a low nadir CD4 count is a well-known factor associated with persistent inflammation and immune activation in ART-treated patients.  In our analysis we did not find any correlation with all the studied parameters.

Reviewer 2 Report

Zuccala P, et al. analyzed multiple immunological and inflammatory parameters before and after DAA in HCV-monoinfected and HIV/HCV coinfected patients. Although the present study was done with relatively small number of patients, they found that immunological parameters were improved after HCV eradication to the similar levels of healthy controls, but not all. The results may be interesting, but several issues should be addressed.

Major

1.     Although authors analyzed many immunological and inflammatory parameters, the conclusion of this study is obscure. Authors should discuss why some parameters did not normalize after DAA, using data. Is it because of liver fibrosis or HIV? These are not clear in the present demonstration.

2.     Authors should clearly describe why patients with HCV-monoinfection and HCV/HIV coinfection were compared and what the results were.  

Minor

1.     Authors should clearly describe that HIV patients were treated with ART or not.

2.     Line 372: Authors should not change the paragraph here.

3.     Line 286: InterestingInterestingly  

Author Response

REVIEWER #2

Major

Comment:

  1. Although authors analyzed many immunological and inflammatory parameters, the conclusion of this study is obscure. Authors should discuss why some parameters did not normalize after DAA, using data. Is it because of liver fibrosis or HIV? These are not clear in the present demonstration.

Response:

We specified in the conclusions that our results suggest that the immunological differences that persist despite the eradication of HCV in both groups could be influenced by the advanced fibrosis in mHCV and to the presence of ongoing HIV replication despite ART in HIV/HCV coinfected patients (lines 593-596).

We summarize all the results in the graphical abstract and made some changes to ameliorate the discussion.

Comment:

  1. Authors should clearly describe why patients with HCV monoinfection and HCV/HIV coinfection were compared and what the results were.

Response:

We compared mHCV with HIV/HCV to evaluate if residual immune activation/depletion in HIV/HCV pts could have an impact on the effects of DAA treatment and the immunological response after eradication of HCV compared to patients without HIV infection.

We found that the effects of DAAs were similar in both groups despite differences in the immunological status and stage of fibrosis at baseline, and that HCV eradication led to sustantial immune recovery in both groups even if some alterations such as increase in plasma MMP-2 and proportion of activated CD8+ T cells persisted.

These sentences have been added in the text to better clarify the aim of the study (lines 64-67 and 73-75).

Minor

Comment:

Authors should clearly describe that HIV patients were treated with ART or not.

Response:

As described in the Session 4.1 “Study population” in Materials and Methods (lines 441-442), being on stable ART for more than 1 year and having undetectable HIV viral load (<20 copies/ml) is a PLWH inclusion criteria.

Comment:

  1. Line 372: Authors should not change the paragraph here.

Response:

We apologize with the reviewer, but we did not find correspondence between the line reported on our manuscript and the one indicated by the reviewer.

Comment:

  1. Line 286: Interesting→Interestingly

Response:

This word has been corrected (line 341).

Reviewer 3 Report

Direct acting antiviral (DAA) therapy has proven its effectiveness in treating HCV infection, however, restoration of the immune system, that was damaged by the infection has not been fully achieved yet. This is due to the fact that sufficient clinical data and studies are not available that can corelate the pre and post treatment immune modifications. The current study in important in this context. The authors here have studied the pre and post DAA treatment circulating cytokines levels and immune markers that are activated during the HCV infection. Although the patient numbers were small, the comparative data were statistically significant and clearly demonstrate lowering of some of the key cytokine levels post treatment. The authors have studied the effect in both HCV alone and HIV/HCV infected patients. The data were ell represented and rational discussion explains a clear picture of how the HCV is interacting with the immune system and then after the VAA treatment, the immune system tends to normalcy. However, some immune component remained unaltered post treatment. The current work has successfully indicated some cytokine markers that have shown significant change in pre and post treatment states, that can serve diagnostic markers. Overall, the information in the current article is important in further understanding the post VAA-treatment immune system alteration in HCV infection. The article discussed the background and results well. Hence, I recommend this article to be published in its current form.

Author Response

We thank the Reviewer for appreciating our study

Round 2

Reviewer 2 Report

Authors answered all questions I asked.